# Evaluation of the Size-of-Source Effect in Thermal Imaging Cameras

**DOI:** 10.3390/s21020607

**Published:** 2021-01-16

**Authors:** Igor Pušnik, Gregor Geršak

**Affiliations:** Faculty of Electrical Engineering, University of Ljubljana, SI-1000 Ljubljana, Slovenia; gregor.gersak@fe.uni-lj.si

**Keywords:** temperature, thermal imaging camera, accuracy, size-of-source effect, SSE, region of interest, ROI

## Abstract

In numerous applications, including current body temperature monitoring in viral pandemic management, thermal imaging cameras are used for quantitative measurements. These require determination of the measurement accuracy (error) and its traceability (measurement uncertainty). Within error estimation, the size-of-source effect (SSE) is an important error source. The SSE is the relation between the physical size of a target and the instrument’s nominal target size. This study presents a direct evaluation of the error due to the SSE. A stable and uniform temperature, generated by blackbodies, was measured by a high-quality thermal imager. To limit the generated radiation, custom-made blocking tiles with different apertures were used. Effects of aperture shapes and positions, camera-target distances and temperature levels on the error were investigated. The study findings suggest that due to the SSE the measured temperatures are too low, especially at longer camera-target distances. The SSE error depends on the number of pixels available and included into the region of interest, for which the accurate measurement is about to be performed. For an accurate temperature measurement, an array of at least 10 × 10 pixels should be exposed to the observed target radiation, while 3 × 3 central pixel area should be included in the temperature calculation.

## 1. Introduction

In the recent years, non-contact temperature measurements using thermal imaging cameras have evolved and many applications have benefited from the progress. Existing thermal imaging cameras are mainly used for qualitative measurements (e.g., identifications of hotspots in technical or medical diagnostic, search for thermal bridges in building efficiency studies), much less for quantitative ones, where accurate absolute values of temperature are required [1,2,3]. In the latter, it is difficult to determine the true temperature of an observed area, called also region-of-interest (ROI), target or source of radiation, because many parameters are affecting the calculation of temperature: radiation magnitude, emissivity of the source, wavelength of the detector measuring the radiation, angle between the camera and the source, distance from the source, ambient temperature, air humidity, background radiation and so forth [4].

One of the basic and simplest counter measures within the current global COVID-19 viral pandemic is the identification of potentially infected people by means of body temperature monitoring. Body temperature measurements being one of the fundamental diagnostic tools of medical science are nowadays performed using classic contact thermometers (e.g., liquid-in-glass or electronic thermometers) or non-contact thermometers (e.g., radiation thermometers like ear or forehead thermometers and thermal imaging cameras).

One of the parameters, which could greatly affect the accuracy of measurement and represents an important uncertainty contribution, is the size-of-source effect (SSE). The SSE is a phenomenon associated with the change in the radiation thermometer output signal due to changes in the size of the source [5,6]. The SSE phenomenon is rather well known and described for classical (industrial) radiation thermometers [7,8], for which there are techniques available for reduction or correction for the SSE [9,10]. Less knowledge and data are available for thermal imaging cameras and are related to either very specific measurement applications [11,12], rely on the techniques, which are much more adequate for radiation thermometers [13] or represent a special design of thermal imaging camera [14]. In [15] an attempt was made how to compensate the SSE in a thermal imager but the technique needs further research and simplification as well as inclusion into a standardized calibration process.

Within body temperature monitoring it is very important to be aware of the relation of the size of detected radiation signal (i.e., output signal of the radiation thermometer) and the size of the radiation source (i.e., target, human face, forehead, eye canthus). Accurate temperature measurements are only possible, if also the effect of this relation is encompassed into the error budget of the radiation thermometer, for example, thermal imaging camera.

This paper presents the design and application of a measuring system for evaluation of the SSE in thermal imaging cameras. The system was used for determination of the SSE error in the evaluation of a high-quality commercial thermal imager.

## 2. Radiation Thermometers and the Size-of-Source Effect

At a certain distance, a radiation thermometer has a given nominal target size depending on the field of view (FOV), which is defined by the optics of thermometer (Figure 1). Theoretically, one could perform a measurement correctly and obtain an accurate result, if the nominal target size of a radiation thermometer is covered entirely by the measured source (target). Due to limitations in the optics, this is not always possible in practice. The phenomenon was named the SSE. The result is an error in measured temperature because the radiation with origin outside the nominal target size influences the measured temperature within the nominal target.

The SSE is a consequence of radiation scattering on dust particles, reflections between lens surfaces, diffractions and aberrations in the optical system of a radiation thermometer. For determination of the SSE one must use a radiation source of stable and uniform spectral radiance with a known diameter. For radiation thermometers with linearized signal and direct reading of temperature the equivalent radiance method [16] can be used, if dealing with a monochromatic radiation thermometer. The SSE is calculated from the radiance *L_λ_*,*_b_*, which is determined by temperature *T* according to Planck’s law:(1)$$L_{\lambda ,b}\left(\lambda ,T\right)=\frac{c_{1L}}{n^{2}\cdot \lambda^{5}}{\left[e^{\frac{c_{2}}{n\lambda T}}-1\right]}^{-1},$$
where *c*_1*L*_ = 2*hc*_0_^2^ = 1.191062·10^−16^ [W·m^2^/sr] is the first and *c*_2_ = *hc*_0_/*k* = 0.014388 [m·K] is the second radiation constant, *λ* is the detection wavelength of the radiation thermometer and *n* is the refraction coefficient of the medium, through which the radiation is travelling. The radiation constants depend on the speed of light in vacuum *c*_0_ = 299,792,458 [m/s], the Planck constant *h* = 6.626176·10^−34^ [J·s] and the Boltzmann constant *k* = 1.380649·10^−23^ [J/K].

Manufacturers of radiation thermometers usually define the nominal target size as a radius or a diameter of a measured circle at a certain distance in which 90% (or up to 98%) of radiation is collected [17]. The nominal target size depends on the type of detector (wavelength) and is usually marked with the viewfinder (Figure 1). For different types of the detector there might be several viewfinders marked. Unfortunately, many radiation thermometers, especially commercial thermometers which operate in the infrared spectrum intended for temperatures below 600 °C, are suffering from a bad SSE characteristic. Based on our experience with the SSE in classical radiation thermometers, we studied the effect of SSE on the overall accuracy of the temperature measurement. Due to the different technology of the imaging camera detector, the methods for SSE determination, as described in [7], are not useful. However, some parts of the proposed measurement system were used—the liquid-cooled holder of radiation-blocking tiles with apertures and sources of radiation, which provide the best possible stability and uniformity of radiated heat flux at certain temperature.

Thermal imaging cameras commonly have a built-in display of various pixel resolution. A pixel represents a temperature point on the 2D thermal map. It is important to note that a certain number of pixels covering the target are required for an accurate measurement [17]. Suggestions are given that the measured area should be covered with minimally an array of 3 × 3 pixels [18,19]. To calculate the maximum suitable distance from the target it is necessary to know the specifications of a thermal imaging camera, such as (i) the resolution of the detector, (ii) its spatial resolution (known also as the instantaneous field of view (IFOV)), (iii) the lens field of view (FOV) and (iv) the absolute size of a measured area. The spatial resolution describes the size of a single pixel (viewing area *a* by *a*) at a given distance (Figure 2).

A camera with a higher resolution is more accurate at a given distance than a camera with the same FOV and lower resolution of the detector. Note that the digital zoom does not improve the accuracy of measurement.

Instantaneous field of view (IFOV) is important for determining the size of an area that a single pixel can detect in terms of the FOV
(2)$$\mathrm{IFOV}/\mathrm{mrad}=\frac{\mathrm{FOV}}{\mathrm{pixel}\;\mathrm{number}}\;\cdot \frac{\pi }{180}\cdot 100,$$
where FOV is the camera’s field of view in degrees (horizontal or vertical), pixel number (horizontal or vertical) is the total number of pixels in a certain direction and π/180·1000 is the conversion factor to mrad units. To calculate IFOV in millimeters one must multiply the IFOV in mrad by the distance *l* in meters
(3)IFOV/mm=IFOV/mrad·l/m.

For example, if the horizontal FOV is 45° and number of pixels 640, IFOV equals 1.23 mrad. Approximately the same number is calculated for the vertical FOV of 34° and 480 pixels. That means this thermal imager at a distance of 1 m has a 1.51 mm^2^ one-pixel area. According to Equation (3), for an accurate reading at 1 m distance a target of at least 13.62 mm^2^ is needed, which corresponds to 3 by 3 pixels. These are the theoretical values while the goal of this study was to determine the influence experimentally.

The technology for detecting radiation is different in classical radiation thermometers compared to thermal imaging cameras. Therefore, the SSE evaluation in this study was performed with a thermal imaging camera. Due to its high resolution, an uncooled microbolometer thermal imager camera with focal plane array (FPA) detector was used in this study. Unlike classical radiation thermometers, thermal imaging cameras have the field-of-view composed of individual detectors or pixels (iFOV). In order to investigate the effects of different shapes of targets, radiation-blocking tiles with different number of apertures of different dimensions and shapes were designed. By blocking the reference radiation from the blackbodies, the tiles enabled a direct evaluation of thermal imager measuring error due to the SSE. That is, tiles were used to limit the heat radiation. We used tiles with slot apertures to investigate the effects of limiting the radiation of a source from two sides, while we used tiles with square apertures for limiting the radiation from four sides. In this way we were able to investigate dependence of the measured temperature from the number of pixels taken into account for its calculation. In addition, the influence of camera-target distance was investigated.

## 3. Material and Methods

### 3.1. Thermal Imaging Camera

A high-quality thermal imaging camera (FLIR T650SC) was used in the study [20]. The detector type was an uncooled microbolometer with the FPA, operating in the spectral range from 7.5 μm to 14 μm, noise equivalent temperature difference NETD < 30 mK and the specified accuracy of ±1% of reading or 1 °C in the temperature range from 5 °C to 120 °C, at ambient temperatures 10 °C to 35 °C. The camera has a wide-angle 45° lens (f = 13.1 mm), with the FOV 45° × 34° mm, spatial resolution 1.23 mrad (IFOV), continuous zoom (8×) and the minimum focus distance 15 cm. Emissivity value can be set in steps of 0.01 from 0.10 to 1.00. Measurements are corrected for the reflected temperature, optics transmission, atmospheric transmission and external optics. Image analysis in the associated software environment (ResearchIR Max by FLIR) enables analysis of different ROIs, for example, spots, areas, automatic hot/cold point detection, difference of temperature, isotherms, line profiles, alarms, temporal temperature dependence and so forth.

The main reason for selection of this camera was its feature to export the temperature value of each individual pixel to a data file. The resolution of the camera is 640 × 480 pixels, which amounts to 307,200 pixel temperatures in a single thermogram. This high resolution is important for a comprehensive analysis of the SSE.

### 3.2. Sources of Heat Radiation—Blackbodies

Stable source of heat radiation (spatial and temporal) was provided by a set of seven blackbodies enabling measurements in the range from 50 °C to 1400 °C. In metrology, a blackbody is a stable and uniform temperature source used for calibration of radiation thermometers [21]. Properties of blackbodies used in this study are shown in Table 1.

Note that some blackbodies were custom designed or substantially modified commercial products. A special blackbody with a large aperture OB XL Piro (Figure 3) was designed specifically for calibration of thermal imaging cameras. Its physical dimensions and metrological specifications enable calibrations of the entire image of a thermal imager [21].

The value and temporal stability of the reference temperature radiated by the blackbodies were monitored using calibrated, traceable reference thermometers. In some cases it could took several hours for a blackbody to reach the stable temperature. To achieve the best accuracy the camera must be stabilized at the operating (room) temperature, therefore it was switched on 30 min before taking the measurements.

### 3.3. Radiation Blocking Tiles

Unlike radiation thermometers, thermal imaging camera FOV consists of individual detectors or pixels (iFOV). To investigate the SSE, custom-made tiles were placed in front of the blackbody aperture in the axis of the camera lens. Tiles were used to physically limit the heat flux radiated from the blackbody measured by a thermal imaging camera. A series of tiles with slots of different dimensions, different number of slots, different distances between the slots and tiles with square apertures were designed (Table 2, Figure 4).

The tiles were made of 2 mm thick aluminum plates with laser-cut slots and square apertures, coated with a special black paint (Pyromark 800) with the emissivity of 0.91, measured in the infrared spectrum of 7 µm to 16 µm. For a simple exchange of different tiles, they were positioned into a customized liquid-cooled holder (Figure 5) [7].

Various apertures in the blocking tiles were designed. Tiles’ shapes, dimensions and aperture positions were coded in the following manner. For example, in the slots tile marked 5(3×)2, the first number represents the slot width in mm (i.e., 5 mm), the number in brackets represents the number of slots in a tile (i.e., three slots) and the last number represents the distance between adjacent slots in mm (i.e., slots are 2 mm apart). Square apertures tiles were coded similarly, for example 5(2 × 2)5, the first number representing side length of the square in mm, the number in brackets the grid of squares and the last number the distance between adjacent squares in mm.

### 3.4. SSE Evaluation Protocol

The holder with tiles was positioned in front of a blackbody as near as the blackbody and its housing allowed. The aim was to reach the highest possible number of pixels in a selected ROI. The camera was positioned at a distance of 10 cm to 100 cm from the tile. The camera, the tile and the blackbody were aligned horizontally so that the blackbody center was in the center of the camera screen (Figure 6).

Part of the study was searching for the relation between the indicated (measured) temperature of the thermal camera and camera position (distance from the tile). In order to avoid errors due to autofocusing of the thermal camera, its focus was set manually to the tile aperture in all measurements. Additionally, some measurements were performed also at the focus distance of 10 cm (camera’s minimum focus distance being 15 cm), because in the process of focusing one could easily make an error of 5 cm or more, if not enough attention is paid. For example, when measuring the temperature of the blackbody at 755.6 °C ± 0.8 °C at the camera distance 10 cm from the tile, the measured temperature was 30 °C higher. According to our experience in calibration of thermal imagers, in some cases, low-quality thermal imagers exhibit similar errors due to wrong focusing and consequently blurred image even at lower temperatures.

The ambient temperature and relative air humidity were recorded and controlled by room air conditioning in order to prevent large fluctuations in ambient conditions. The recorded ambient conditions (20 °C to 23 °C, 30% rh to 50% rh) were input parameters in the thermal imager data processing software ResearchIR Max.

### 3.5. Thermogram Analysis

Rectangular shaped ROI, positioned at the border of the physical aperture was used as shown in Figure 7. The analysis was performed with the ResearchIR Max software (Figure 8).

For reliable measurements it is important that the entire tile aperture is covered by a stable temperature source. For example, in the bottom line of the squared apertures in Figure 8 the stable temperature source was not available along the entire line. Due to physical limitations such as the size of the blackbody aperture and the tile distance from the blackbody, we decided on the most suitable tiles for the main measurements using shorter tile-camera distances. If the distance was too large, not enough pixels were available for temperature determination in the selected ROI. Therefore, measurements of certain tiles were only performed at shorter distances.

### 3.6. Data Processing

Thermal imaging cameras were evaluated using a very stable radiation source—the blackbody. Because a series of temperature measurements was performed, the following naming is used: (i) measured temperature *T*—indication of the thermal imaging camera, (ii) reference temperature *T_ref_*—temperature due to the radiated heat flux from the blackbody. The error of the camera compared to the temperature of a blackbody was calculated. Camera’s relative measurement error *E* in % was calculated according to Equation (4).
(4)$$E\;=\;\frac{T\;-\;Tref}{Tref}/\;\%.$$

To evaluate the time stability of indicated temperature *T* the spread of the measurements was used. A peak-to-peak factor Δ in % was introduced, presenting the maximal difference in a time series of measurement (Equation (5))
Δ = (*max* (*T*) − *min* (*T*))/*T*_ref_/%(5)
where *max* (*T*) and *min* (*T*) are the maximal and minimal temperatures *T* measured by the thermal imaging camera at a certain temperature point *T*_ref_. Factor Δ represents the stability of the measured temperature; the smaller its value the more repeatable the indication of the thermal imaging camera is.

Temperature measurements of the tiles with single or multiple slots were performed and the results analyzed. In addition, thermograms of tiles with different shapes of apertures were studied. With multiple square aperture tiles, the temperature profile was calculated along the line, thus forming the temperature gradient of the line. Special attention was given to the analysis of ROI regions in vicinity of the aperture edge.

Measurements with the ROI placed along the edge of a single slot aperture were performed. The vicinity of the aperture edge was hypothesized as a critical region for calculation of temperature in a thermal image, especially if there was a larger temperature gradient between the ROI and the neighboring area. The objective was to determine number of thermogram pixels from the border of the ROI which were actually influenced by the SSE. Measurements were performed at various temperatures from 50 °C to 1400 °C.

## 4. Results

### 4.1. Single Slot Tiles

For accurate measurement, the camera manufacturer recommends a ROI of at least 3 × 3 pixels [22,23]. In our study thermal images at longer distances and using tiles with the narrowest slots, the slot width did not suffice for a 3 × 3 pixels area. The measurements with ROI of one or two columns of pixels were in-accurate, for example, at reference temperature of 755 °C the average camera temperature was 661 °C. Figure 9 shows limited number of pixels, which were covering the measured area.

By decreasing the slot width, the average measured temperature decreased at all distances and temperatures. Figure 10 shows temperatures measured at different distances and using two blackbodies with very different apertures (diameter 60 mm and 263 mm (XL)) at 50 °C. Using larger blackbody aperture resulted in lower error. Additionally, the error decreased with the slot width (Figure 10). The results show that at the distances of up to 20 cm the measured temperatures were even higher than the temperature of the blackbody. Similar results were found at 1000 °C for blackbodies with 40 mm and 50 mm apertures (Figure 11).

Figure 12 shows the blackbody temperature and tile aperture shape dependence of the Δ (Equation (5)). The Δ increased with the decreased slot width and for almost all blackbodies with the increased temperatures, an exception being XL blackbody with the largest aperture at 50 °C XL. This could be explained with the fact, that only the very large aperture XL blackbody enabled uniform temperature source for all the apertures. Smaller number of available pixels in ROI increased the error *E* (Figure 12). For example at 3 mm slot width and reference temperature 1000 °C the error was −147 °C, which is roughly Δ = 15% and dependent on the camera-tile distance. Even at the largest slot width of 40 mm and 1000 °C the error was −63 °C (Δ = 6%) which is far beyond the specified accuracy of 2%.

Although the manufacturer recommends the ROI of at least 3 × 3 pixels for accurate measurements [23], our measurements showed that at any temperature larger number of pixels are necessary, at least 10 × 10, suggested by the same manufacturer as well but not specified how many pixels are taken into account for calculation of temperature [24].

### 4.2. Multiple Slot Tiles

To further analyze the influence of the SSE we developed the tiles with multiple slots, different width of slots and different distance between the slots. The error of the thermal imaging camera in % at different temperatures and tiles with multiple slots at the distance 10 cm is presented in Figure 13. The distance of 10 cm was the only one at which we were able to perform measurements of all tiles with multiple slots in the complete range of temperatures because not enough pixels could cover the narrow slots at larger distances.

The error decreased with the smaller width of slots and smaller distance between them. The error typically increased at higher temperatures. The results at 500 °C were different for the narrowest slots of 1 mm because the distance between the tile and the aperture of the blackbody (the cesium heat pipe was placed 15 cm into the furnace, Figure 6 right) is larger than for other blackbodies. Therefore, the pixel area at the aperture of the blackbody was larger than in other cases, which increased the influence of the SSE. The error is practically the same, if the width of slots is the same, regardless of the number of slots (3 or 5).

### 4.3. Thermogram Analysis of Tiles with Different Apertures Shape

The ROI compared to the camera FOV can range from a few pixels to a few thousand pixels (Table 3), that is, from a few percent to a few thousandths of a percent. The sensor of the commercial imager used (T650sc by FLIR Systems) has 307.200 pixels in a 640 × 480 array. The ROI used was the square and the same ROI was used to calculate the average temperature of slots. The comparison of the average temperature of slots and squares showed the higher values of slots versus squares, which is the consequence of the SSE. Namely, the source of radiation is limited more with squares than with slots, using the same ROI.

Further evaluation of the SSE was performed by comparison of average temperatures at the same dimensions of selected slots and squares, which were measured almost at the same time and under the same conditions (Figure 14). For the same slot and square dimensions, the same number of pixels or ROI of the same size was used. The specified accuracy of thermal camera FLIR T650sc at 250 °C is ±2%. The reference temperature was chosen as the measurement that included the maximum number of pixels with the maximum measured temperature (slot width 15 mm).

Figure 15 shows an example of a thermal image at 250 °C with a single square aperture tile 10(1 × 1).

With the decreased number of pixels in ROI, the differences between the measured temperatures of slots and squares increased. Figure 16 shows average differences between temperature of slots and squares of the same width. Additionally, the differences increased further as the reference temperature increased. For example, at 1000 °C the average temperature difference across all distances was as high as 32.3 °C at 3 mm aperture width6.

The temperature difference increased with the camera-tile distance due to decreased number of pixels in calculation of the average temperature. An example is shown in Figure 17 for the same width of squares 10(2 × 2)10, 5(2 × 2)10, 3(2 × 2)10 and slots 10(1×), 5(1×), 3(1×) at distances of 10 cm and 20 cm and different temperatures.

Figure 18 and Figure 19 present the average measured temperature of slots and squares of the same dimensions at distances 10 cm and 20 cm, at 500 °C and 1000 °C, respectively. Temperature differences between slots and squares of the same dimensions at the distance were in line with the specified accuracy (±2%) only at 500 °C. At the same distance the temperature reading of the ROI is higher for the slot compared to the square of the same dimension because the square blocks the radiation from both directions (horizontal and vertical). Temperature decreased for the slots with smaller width (2nd, 4th, 6th measurement) and for the squares with smaller dimensions (1st, 3rd, 5th measurement).

### 4.4. Thermogram Analysis of Complex Tiles

Using square apertures in complex 3(3 × 3)3 blocking tiles in front of the cesium heat-pipe blackbody at 500 °C (499.6 °C), the temperatures measured by the thermal imager were analyzed. The red lines in Figure 20 represent lines of pixels in the temperature profile, that is, spatial gradient of the line shown in Figure 21, Figure 22 and Figure 23. Note the difference in temperature source for individual apertures in Figure 20 (left) caused by an insufficiently large blackbody aperture (e.g., aperture was too small to enable a uniform temperature for every square aperture). The phenomenon is clearly visible also in Figure 21 (i.e., different temperatures in individual square apertures).

At the camera-tile distance of 15 cm, the 8 pixels in the middle of the left square were sufficient to determine the temperature with the error of −1.0 °C and the standard deviation of 0.6 °C. If we take into account all pixels along the line inside the square (22 pixels, 7 additional pixels on each side from the middle 8 pixels), the error is −74.2 °C and the standard deviation is 120.7 °C. This shows clearly how important is the knowledge of the SSE in thermal imagers and how many adjacent pixels are influencing the result.

At the distance of 40 cm, the number of pixels decreased to only one and the resulting temperature had an error of −4.3 °C with the standard deviation 11.4 °C (taking into calculation 3 pixels).

At the distance of 100 cm temperature measurement was very inaccurate (the maximum temperature of all pixels in the left square was 473.6 °C, for example, the error was −26.0 °C) with the standard deviation 73.3 °C (taking into calculation 3 pixels). Temperature graph of individual pixels are shown in Figure 21, Figure 22 and Figure 23, while calculation is summarized in Table 4.

### 4.5. Thermogram Analysis of Regions in the Vicinity of the Aperture Edge

Measurements with the ROI, positioned along the edge of a single slot aperture, were performed. Figure 24 shows temperatures of a line of 15 pixels from the edge of the aperture at 50 °C in front of two blackbodies with different aperture diameters of 60 mm and 263 mm (XL), respectively. Pixel no. 15 is positioned on the edge as in Figure 7, therefore it was not considered as a measuring pixel in the slot.

A detailed graph is presented in Figure 25 for temperatures accurate within the specified accuracy of the thermal imager (1% or 1 °C). To reach the specified accuracy it is important to include only the pixels positioned at least seven pixels away from the border of the ROI for temperature calculations, that is, only pixels 1 to 7 (except at 100 cm and the blackbody with smaller aperture).

Additionally, high temperature measurements were performed at 1000 °C, which was selected due to availability of two blackbodies with different aperture diameters. Figure 26 shows temperatures of 15 pixels from the edge of a single slot aperture at 1000 °C in front of blackbodies with aperture diameters of 40 mm and 50 mm (L), respectively.

Figure 27 shows temperatures of pixels, which were at least 7 pixels from the border of the ROI.

The specified accuracy of 2% was not achieved in all cases. Temperatures measured 10 cm from the slot were more inaccurate than other measurements. The reason is non-optimal focusing of the thermal imager, that is, 10 cm is below the thermal imager minimum focus distance of 15 cm. Temperatures measured 100 cm from the slot are inaccurate due to large iFOV which requires larger area with homogeneous temperature to obtain results within the specification of the thermal imager.

## 5. Discussion

The influence of the SSE is one of the parameters that affects both quantitative and qualitative temperature measurements performed by thermal imaging cameras. The SSE should always be evaluated in order to get accurate temperature with a known measurement uncertainty.

Although the use of thermal imagers for body temperature monitoring is rapidly increasing, there is still no widely accepted recommendations or an adopted international standard providing guidelines for greater reliability and accuracy of measurements using thermal imaging cameras. Some attempts are in the process, for example, IEC standard for medical imagers [24].

In order to directly evaluate the SSE we measured radiation of a blackbody radiator partly covered by a series of blocking tiles with slots and square apertures of different dimensions, number and mutual distance. Our study results suggest the SSE is the reason that on a single-slot tile the thermal imager displays lower temperature when reducing the slot width and when increasing the distance between the camera and the tile.

For multi-slot tiles with different distances between individual slots, the measured temperatures increased with the decreased distance between slots. Only at the distance of 15 cm we were able to evaluate all selected tiles but we speculate that the trend of decreasing temperatures at other distances would be similar.

Comparison of temperatures when using the tiles with squares and slots showed the pronounced influence of the SSE on the quantitative temperature measurement with thermal imaging cameras. By limiting the heat radiation of a source from two sides (slot apertures) or four sides (square apertures), we observed differences in the measured temperature. These differences increased with decreased number of pixels, which were taken into account for calculation of the average ROI temperature. At higher temperatures, larger differences were obtained in measurements. In our study, the majority of the performed measurements at 250 °C (roughly 70%, Figure 14) exhibited errors larger than the specified accuracy of the tested commercial thermal imaging camera, although it was one of the best measuring instruments in its class. Measurements of the temperature profile in the square aperture 3(3 × 3)3 at three different camera-tile distances showed that the number of pixels, that could be used for calculation of the average temperature, greatly decreased with the increased distance.

If an accurate (quantitative) temperature measurement with a thermal imaging camera is required, we suggest that the SSE must be taken into account thoroughly. Although manufacturers of thermal imagers recommend that the measured ROI should be covered with an array of at least 3 × 3 pixels, our study showed that for the specified accuracy the measured area, providing it has a homogeneous temperature, should be covered by at least 10 × 10 pixels, of which we should take into account only the central 3 × 3 pixels for the final result. This means that at least seven pixels from the border of the ROI should be excluded from the calculation of the temperature. That is valid for high-resolution and high-quality thermal imagers. Thermal imagers with low-resolution exhibit worse SSE.

If the measured target cannot be covered by an ROI of at least 10 × 10 pixels, the accuracy of the quantitative result is questionable and uncertainty of the measurement is likely larger than the accuracy stated in producer’s specifications.

## 6. Conclusions

Knowledge of thermal imaging camera accuracy in the scientific literature is limited, poorly investigated and rarely questioned. Thermal imaging cameras suffer from the size-of-source effect (SSE) which is far from negligible. Therefore, the accuracy of a thermal imager heavily depends on the SSE, which is not specified by a manufacturer. This fact becomes extremely important when the accurate temperature measurement is required at a larger distance and a small target with non-homogeneous temperature. The latter being for example mass thermal imager temperature screening of people at entry points to detect individuals with elevated body temperature, which might help in mitigation of spreading a contagious disease.

The results of our study suggest that, if reliable and accurate thermal imager temperature measurements are needed, the operator shall take into account at least the following: (i) the size-of-source effect is an important error source of any thermal imaging camera, (ii) area of the ROI shall contain at least 10 × 10 pixels of homogeneous temperature of which the area taken into account for calculation of temperature shall be positioned at least 7 pixels from the edge of the ROI, (iii) depending on the thermal imaging camera lens system the camera-source distance is an important parameter which limits the possibility of performing accurate temperature measurements and (iv) precise focusing is very important for accurate temperature measurements.

## Figures and Tables

**Figure 1 sensors-21-00607-f001:**
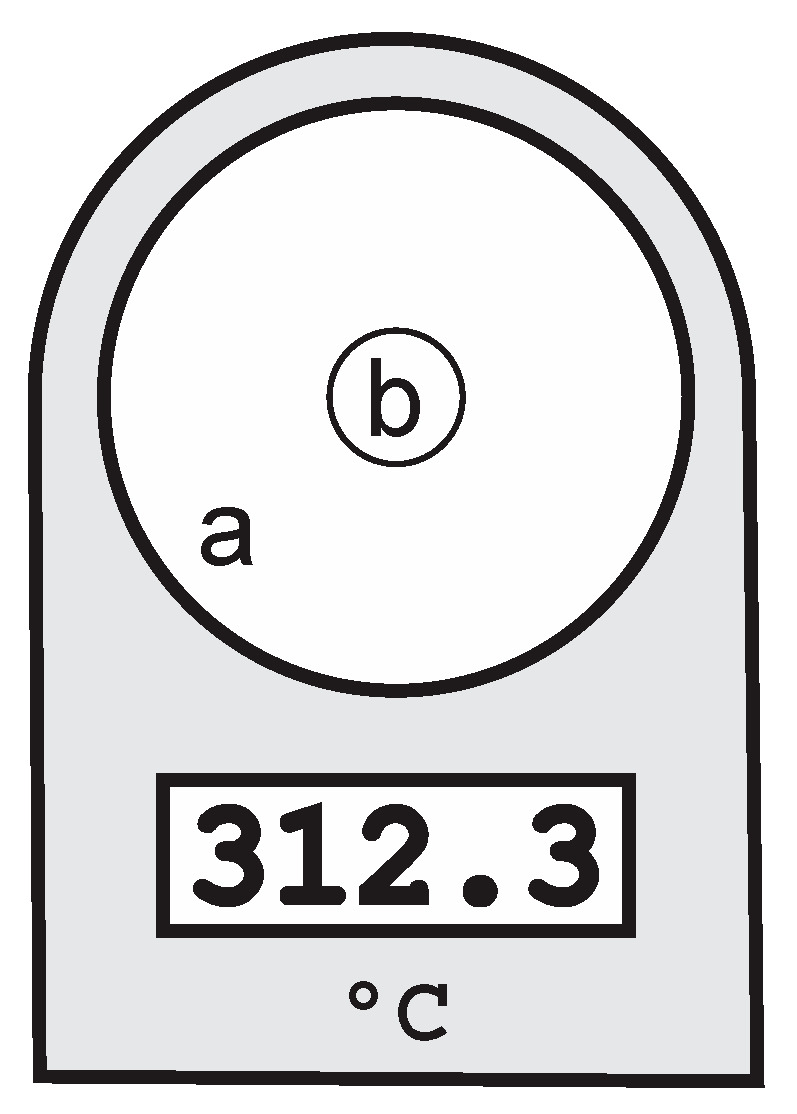
A radiation thermometer with the field of view (**a**) and the nominal target (**b**).

**Figure 2 sensors-21-00607-f002:**
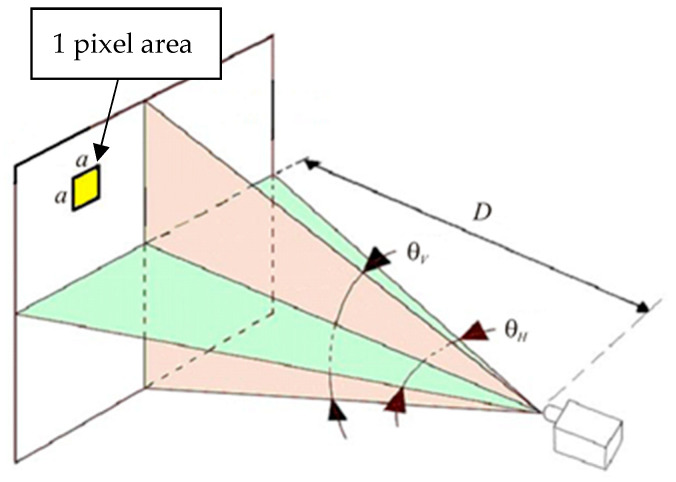
Instantaneous field of view (IFOV) (*a*·*a*) and field of view (FOV) (θ*_H_* and θ*_V_*).

**Figure 3 sensors-21-00607-f003:**
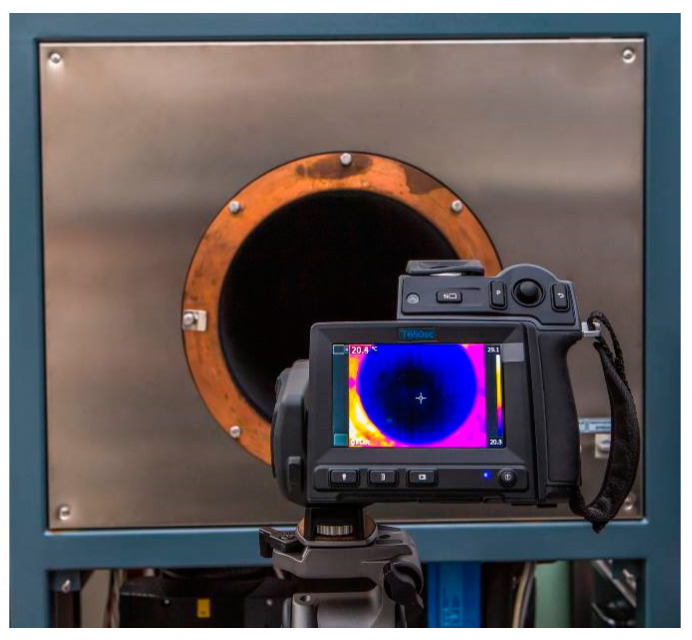
Large aperture blackbody for thermal imaging camera calibrations (OB XL Piro by Kambič).

**Figure 4 sensors-21-00607-f004:**
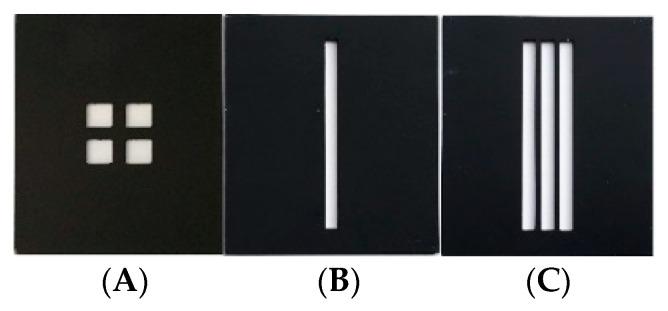
Tiles with apertures of different shapes; squares (**A**) and slots (**B**,**C**). Tiles shown here are coded as 5(2 × 2)2, 2(1×) and 2(3×)1 (**A**–**C**).

**Figure 5 sensors-21-00607-f005:**
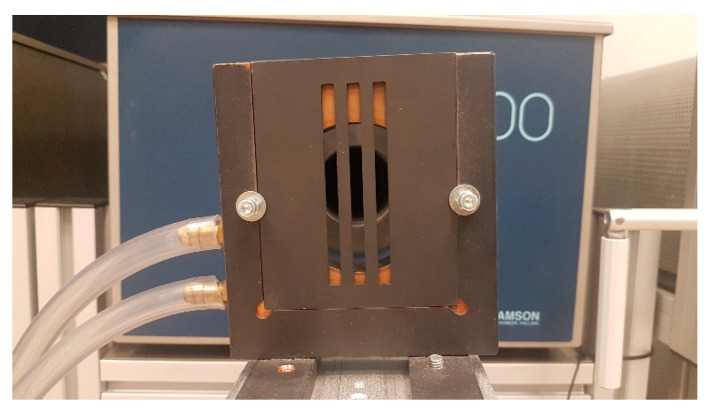
Radiation blocking tiles (dark grey) were fixated into a holder (copper).

**Figure 6 sensors-21-00607-f006:**
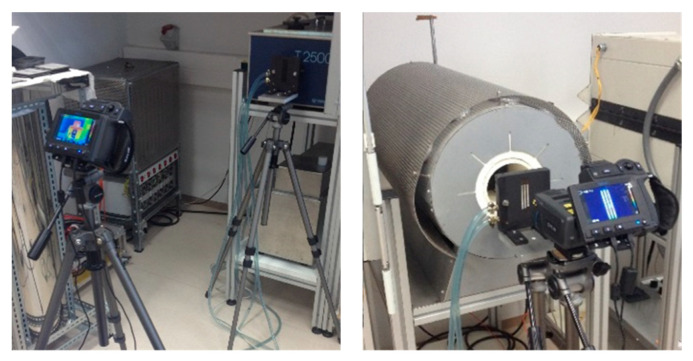
Thermal imaging camera at different distances from the tile—100 cm (**left**) and 10 cm (**right**).

**Figure 7 sensors-21-00607-f007:**
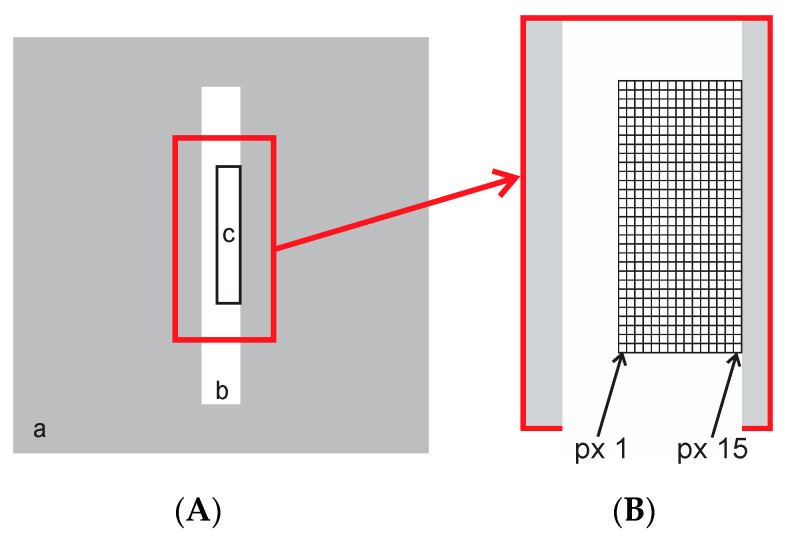
(**A**)—Schematics of the radiation-blocking tile (a) with the slot shaped aperture (b) and the position of the selected rectangular region of interest (ROI) (c). (**B**)—Positioning of a 15-pixel-width ROI along the aperture edge.

**Figure 8 sensors-21-00607-f008:**
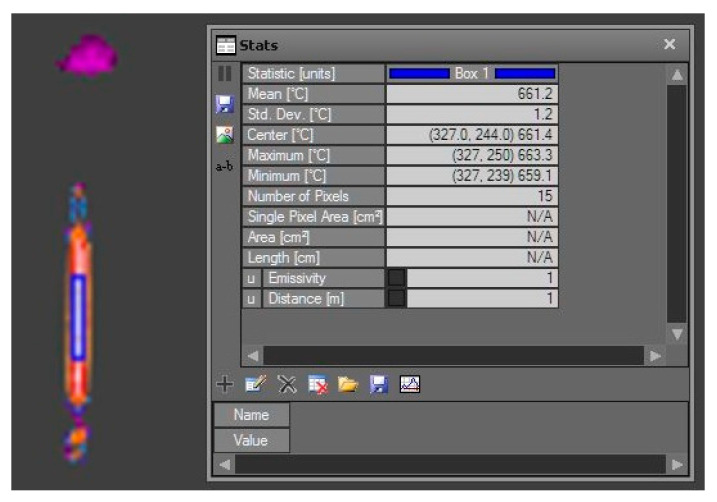
Temperature analysis (**right**) of a 15-pixel-width ROI in a 3(1×) single slot tile (blue rectangular on the **left**) at 755 °C at a distance of 100 cm from the camera.

**Figure 9 sensors-21-00607-f009:**
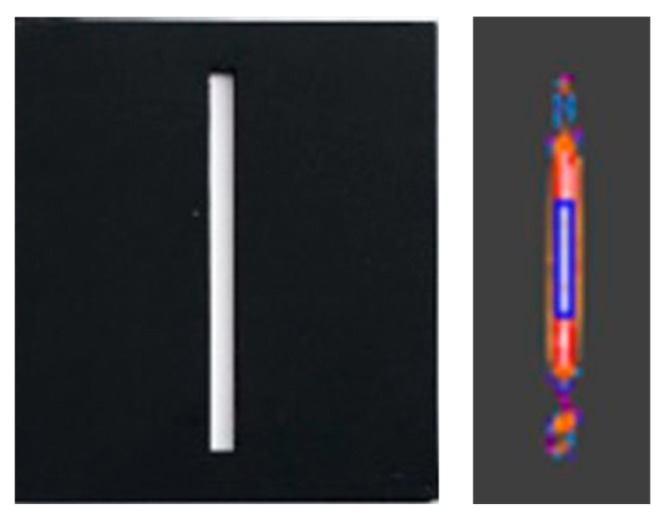
Image of the slot 3(1×) tile in visible range (**left**) and its thermal image at 755 °C at 100 cm camera-tile distance (**right**). On the right, blue rectangular indicates the ROI—a single column of 15 pixels width.

**Figure 10 sensors-21-00607-f010:**
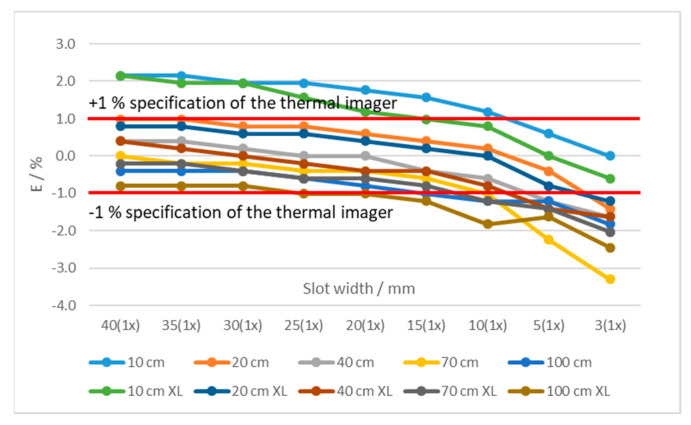
Errors of the thermal imaging camera in % at blackbody temperatures of 50 °C at different camera-tile distances (10 to 100 cm) and using tiles with different slot widths. Two blackbodies were used:60 mm and 263 mm (XL) aperture. Each graph point represents an average temperature of the selected ROI.

**Figure 11 sensors-21-00607-f011:**
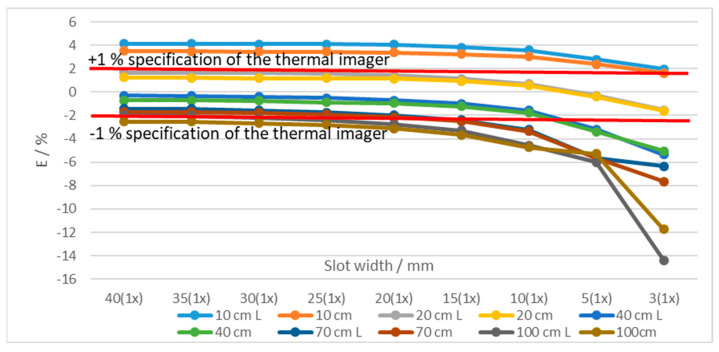
Error of the thermal imaging camera in % at blackbody temperatures of 1000 °C at different camera-tile distances (10 to 100 cm) and using tiles with different slot widths. Two blackbodies were used:40 mm and 50 mm (L) aperture. Each graph point represents the average temperature of the selected ROI.

**Figure 12 sensors-21-00607-f012:**
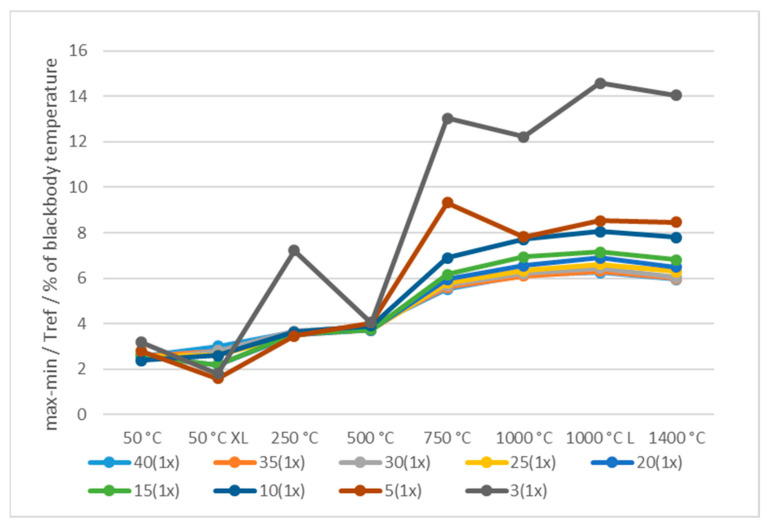
Factor Δ depends on the blackbody temperature and the width of the tile slot.

**Figure 13 sensors-21-00607-f013:**
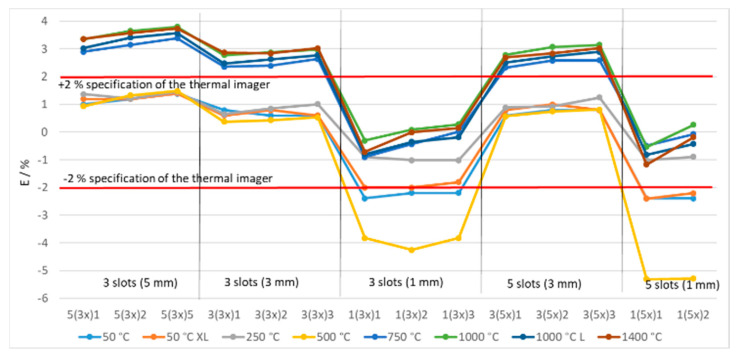
Error of the thermal imaging camera at different blackbody temperatures at different camera-tile distance 10 cm and using tiles with multiple slots. Each graph point represents the average temperature of the selected ROI.

**Figure 14 sensors-21-00607-f014:**
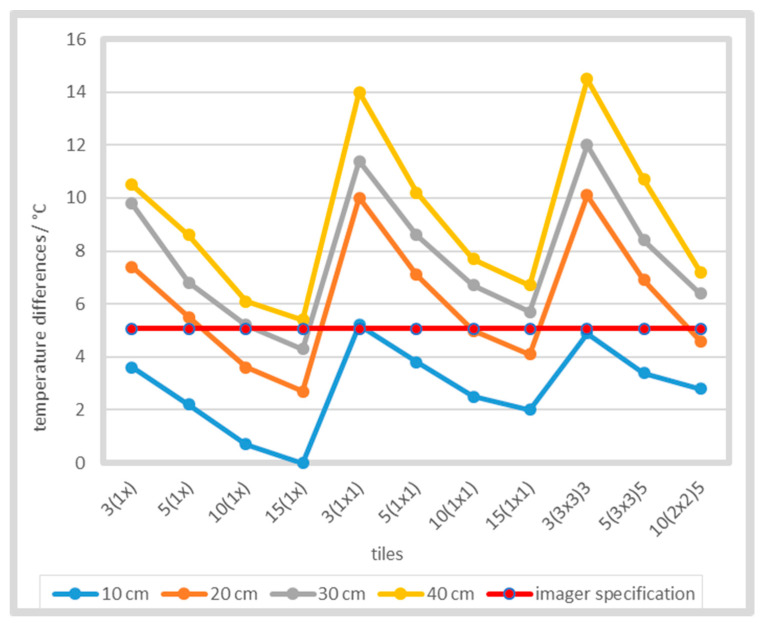
Differences of measured temperatures from the maximum temperature compared to the imager specifications (±2%) at 250 °C and different distances. Each graph point represents the average temperature of the selected ROI.

**Figure 15 sensors-21-00607-f015:**
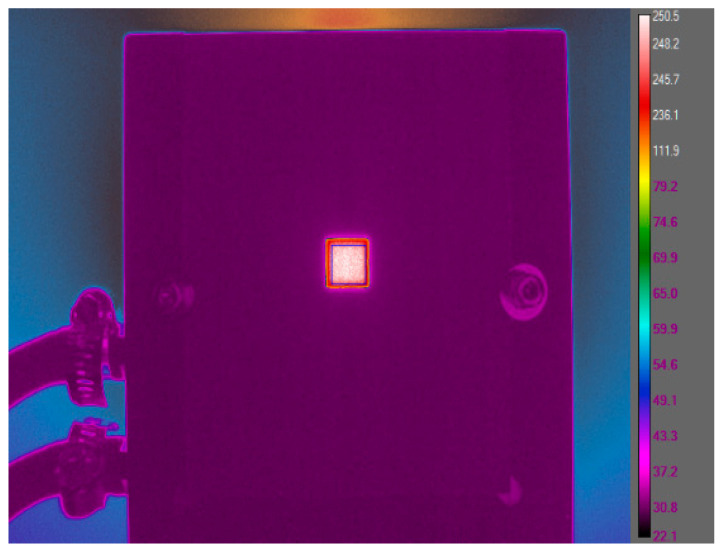
Thermal image of a single square blocking tile at 250 °C. The ROI consisted of 1156 pixels (0.38% of camera’s FOV) for camera-tile distance 20 cm.

**Figure 16 sensors-21-00607-f016:**
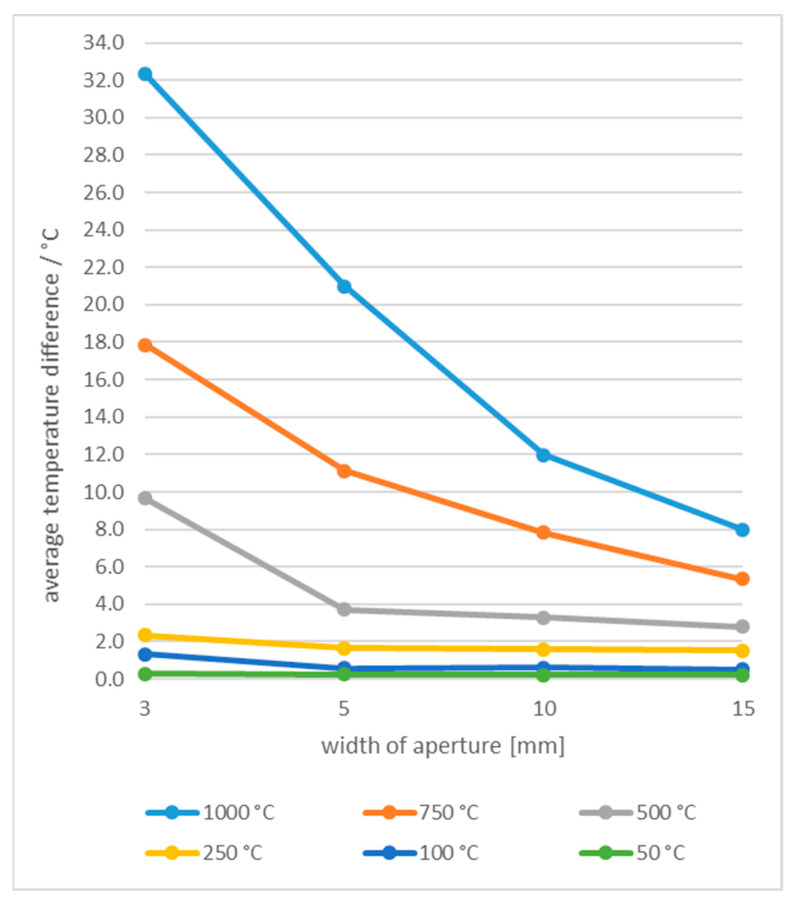
Shape-dependent temperature differences when using slots and squares of the same width.

**Figure 17 sensors-21-00607-f017:**
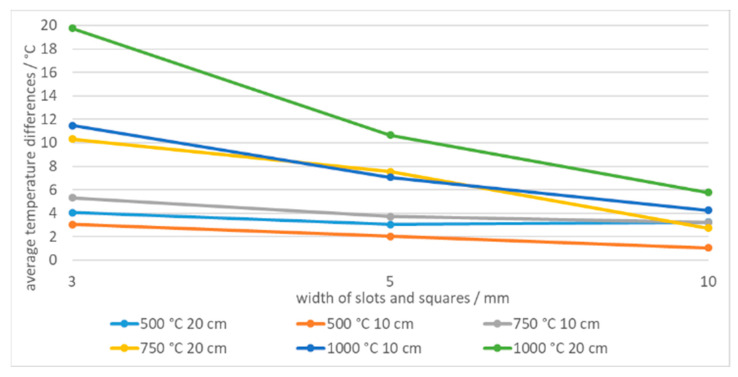
Shape-dependent temperature differences at different camera-tile distances and temperatures.

**Figure 18 sensors-21-00607-f018:**
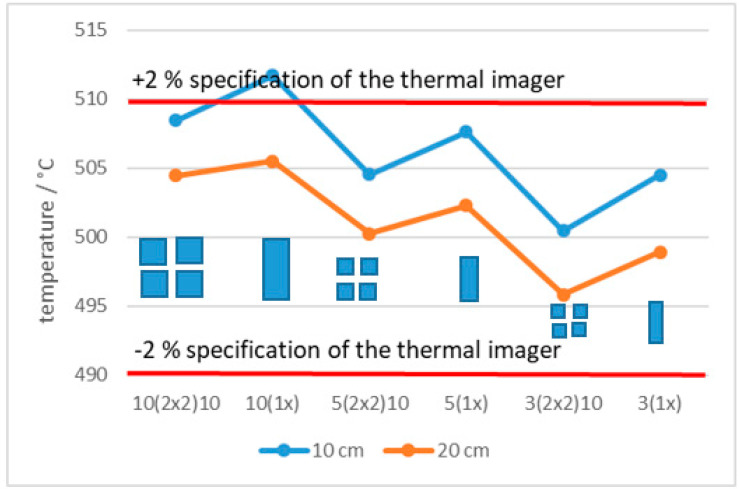
Temperature of slots and squares of the same dimensions at 500 °C.

**Figure 19 sensors-21-00607-f019:**
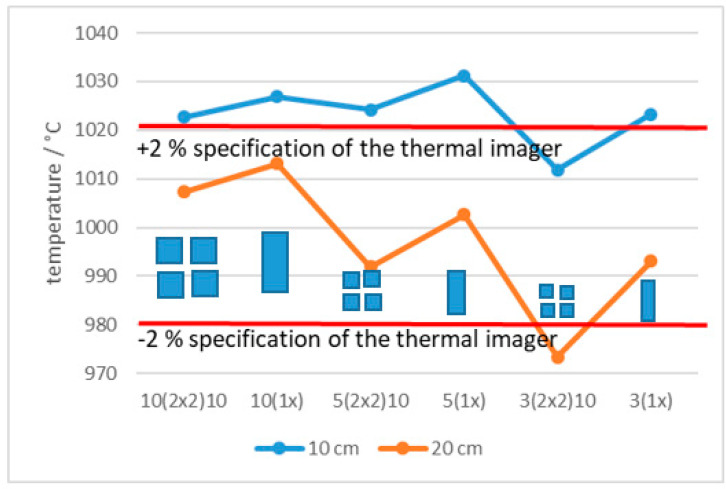
Temperature of different slots and squares of the same dimensions at 1000 °C.

**Figure 20 sensors-21-00607-f020:**
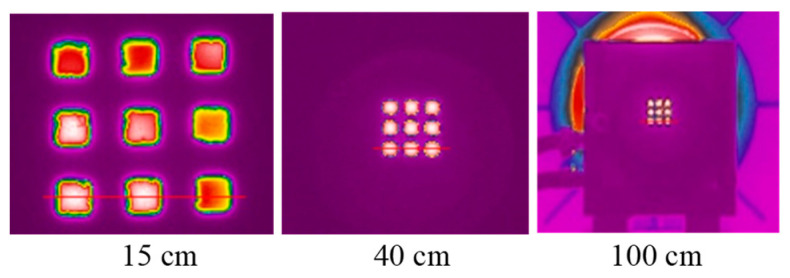
Three thermal images of tile 3(3 × 3)3 at different camera-tile distances. The red line indicates the direction of the temperature profile determination.

**Figure 21 sensors-21-00607-f021:**
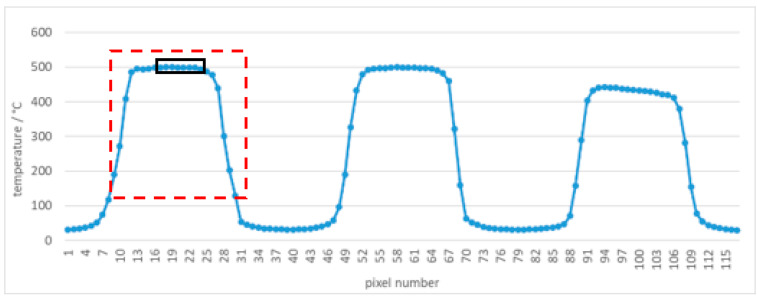
Tile 3(3 × 3)3 temperature profile at 500 °C and the camera-tile distance 15 cm. Red square indicates the tile aperture covered by 22 pixels. Black rectangle indicates the middle 8 pixels of the square tile.

**Figure 22 sensors-21-00607-f022:**
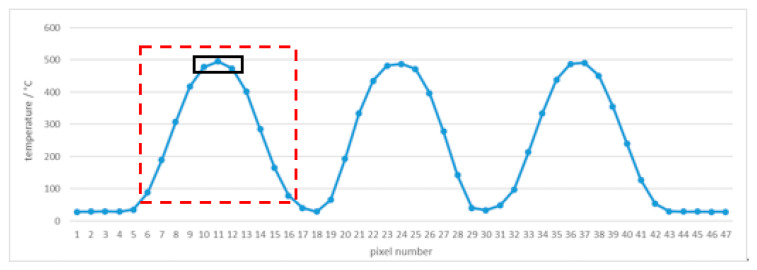
Tile 3(3 × 3)3 temperature profile at 500 °C and the camera-tile distance 40 cm. Red square indicates the tile aperture covered by 11 pixels. Black rectangle indicates the middle 3 pixels of the square tile.

**Figure 23 sensors-21-00607-f023:**
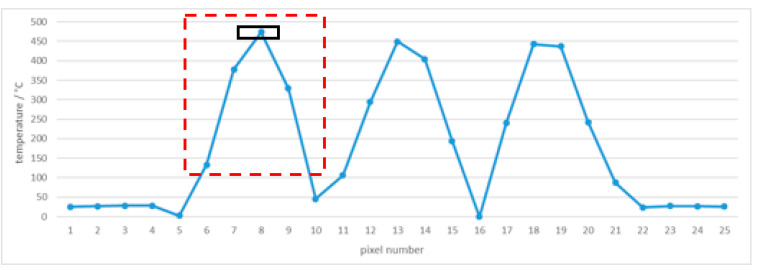
Tile 3(3 × 3)3 temperature profile at 500 °C and the camera-tile distance 100 cm. Red square indicates the tile aperture, covered by only 4 pixel. Black rectangle indicates the middle pixel of the square tile.

**Figure 24 sensors-21-00607-f024:**
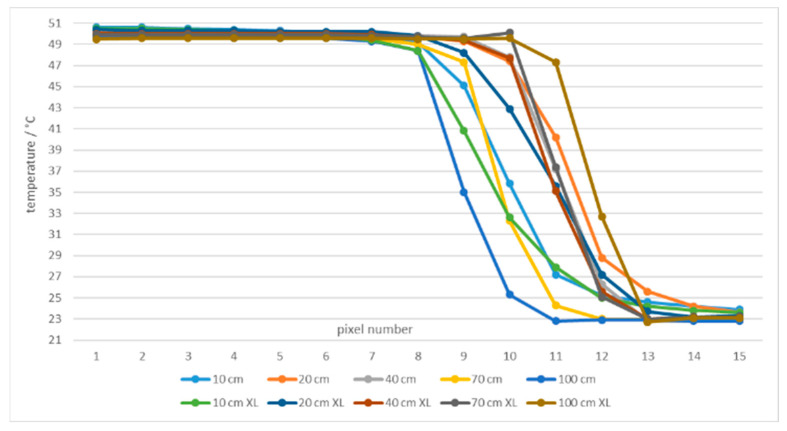
Temperatures of a line of 15 pixels closed to the border of ROI at 50 °C and different blackbodies.

**Figure 25 sensors-21-00607-f025:**
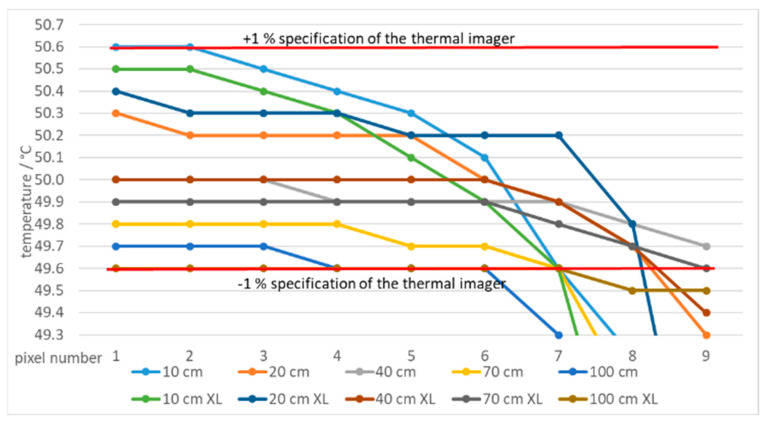
Temperatures close to the border of ROI (15 to 9 pixels from the border) at 50.1 °C and blackbodies with apertures 60 mm and 263 mm (XL). Red lines indicate the thermal imager accuracy.

**Figure 26 sensors-21-00607-f026:**
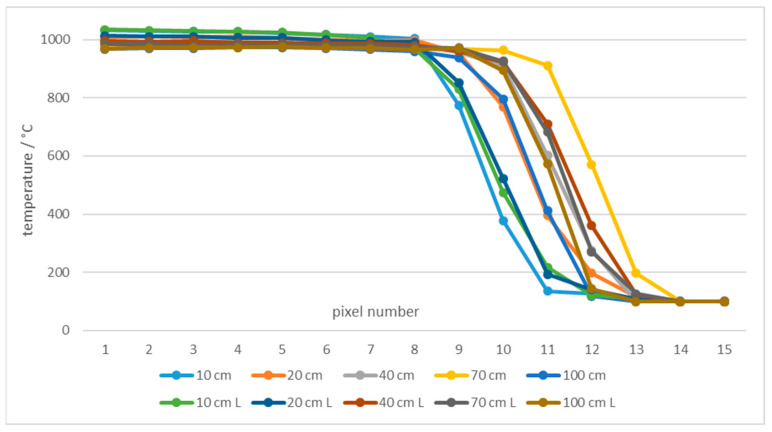
Temperatures of a line of 15 pixels close to the border of ROI at 1000 °C and different blackbodies.

**Figure 27 sensors-21-00607-f027:**
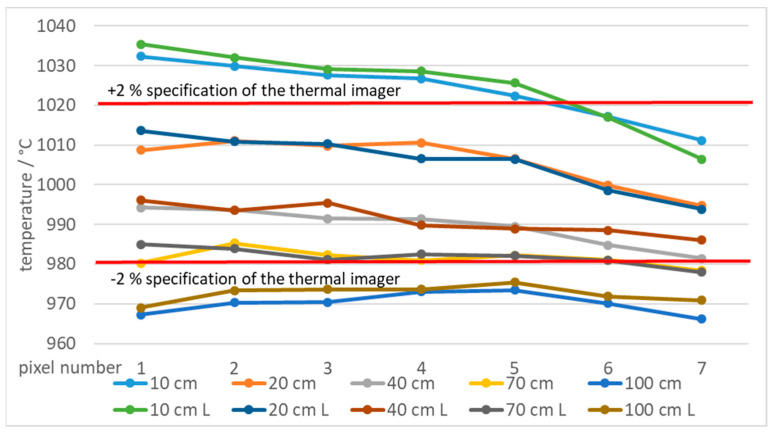
Measured temperatures 15 to 9 pixels away from the border of ROI at 1000 °C and different blackbodies.

**Table 1 sensors-21-00607-t001:** Properties of blackbodies.

Blackbody	Aperture Diameter × Cavity Length (mm)	Thermal Media, Principle	Range (°C)
Land P1600B (L) (by Land Infrared)	50 × 320	SiC furnace	600 to 1500
Na heat-pipe (by ACT)	40 × 500	heat pipe	600 to 1000
Cs heat-pipe (by IKE Stuttgart)	55 × 500	heat pipe	250 to 600
T 2500 (by Tamson)	60 × 400	silicon oil	150 to 250
T 2500 (by Tamson)	60 × 400	silicon oil	60 to 150
T 2500 (by Tamson)	60 × 400	water	5 to 60
OB XL Piro (by Kambič)	263 × 1000	water	10 to 70

**Table 2 sensors-21-00607-t002:** Tiles used in the study.

Single	40(1×), 35(1×), 30(1×), 25(1×), 20(1×), 15(1×),				
slot	10(1×), 5(1×), 3(1×)				
Multiple	5(3×)5, 5(3×)2, 5(3×)1, 3(3×)3, 3(3×)2, 3(3×)1,				
slots	1(3×)3, 1(3×)2, 1(3×)1, 3(5×)3, 3(5×)2, 3(5×)1,				
	1(5×)2, 1(5×)1, 1(10×)2, 1(10×)1				
Square	10(2 × 2)10, 10(2 × 2)5, 5(2 × 2)10,				3(2 × 2)10,
apertures	5(3 × 3)5,	3(3 × 3)5,	3(3 × 3)3,	15(1 × 1),	10(1 × 1),
	5(1 × 1), 3(1 × 1)				

**Table 3 sensors-21-00607-t003:** Number of pixels in the ROI at 250 °C.

Number of Pixels in ROI	Camera-Tile Distance (cm)			
Tile	10	20	30	40
3(1×); 3(1 × 1); 3(3 × 3)3	225	64	36	16
5(1×); 5(1 × 1); 5(3 × 3)5	784	225	121	64
10(1×); 10(1 × 1); 10(2 × 2)5	4225	1156	529	289
15(1×); 15(1 × 1)	10,404	2704	1296	729

**Table 4 sensors-21-00607-t004:** Average values of selected pixels, standard deviations and errors at 499.6 °C.

Distance/cm	Average/°C/No. of Pixels	Standard Deviation/°C	Error/°C
15	498.6/8 425.4/22	0.6 120.7	−1.0 −74.2
40	495.3/3	11.4	−4.3
100	473.6/1	73.3 *	−26.0

* only 3 pixels could be used to calculate the standard deviation.

## Data Availability

The data presented in this study are available on request from the corresponding author.

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
