# Peer review of "Evaluation of the Size-of-Source Effect in Thermal Imaging Cameras"

_sensors, 2021, doi:10.3390/s21020607_

Round 1

Reviewer 1 Report

I found the manuscript is well written and useful for the temperature retrieval from thermal imaging camera, but there are still some concerns need to be responded:

  1. Although the authors have showed the blackbody calibration in the paper, the major problem is still the radiation calibration accuracy for the thermal imaging camera. How well could it meet the requirement of different researches?
  2. Is there any effect of different spectral range on the temperature accuracy for different temperature range, such as a body temperature or 1000 degree? Since the body temperature and 1000 degree have different energy center. Why choose 250 degree, 500 degree and 1000 degree? Why not the body temperature range?
  3. What’s the effect of different emissivity?
  4. Line 79: The subscript of the letters should pay more attention.

Author Response

I found the manuscript is well written and useful for the temperature retrieval from thermal imaging camera, but there are still some concerns need to be responded:

  1. Although the authors have showed the blackbody calibration in the paper, the major problem is still the radiation calibration accuracy for the thermal imaging camera. How well could it meet the requirement of different researches?

The answer to this question is related to the third question. For accurate non-contact temperature measurements, it is essential that we know the emissivity of a measured object (target) and set the same emissivity to the radiation temperature measuring instrument. In addition, several uncertainty contributions make the non-contact temperature measurements less accurate than contact temperature measurements. One important uncertainty contribution is the size-of-source effect (SSE) which is relatively well described for radiation thermometers, at least for the more sophisticated ones. Low cost radiation thermometers have a fixed focus system and bad SSE, which is rarely evaluated because of the price of calibration. For thermal imaging cameras there is a lack of literature in this field. Also the detector technology (matrix of sensors or FPA) can be completely different and therefore the SSE cannot be verified in the same way as for radiation thermometers. On top of that the raw signal for a thermal imager is not available at all (rarely also for a radiation thermometer) and only the best cameras provide an opportunity to acquire the temperature value of each detector element (pixel), which is the prerequisite for evaluation of the SSE. We are convinced that awareness of the SSE in thermal imaging camera is of utmost importance for those applications where accurate quantitative measurements are required and targets are small or/and temperature is important at the border/edge of a measured object.

  1. Is there any effect of different spectral range on the temperature accuracy for different temperature range, such as a body temperature or 1000 degree? Since the body temperature and 1000 degree have different energy center. Why choose 250 degree, 500 degree and 1000 degree? Why not the body temperature range?

The research started prior the current Covid-19 crisis. The pandemic only highlighted the problem of accuracy of thermal imaging in the cases of body temperature measurement and the importance of accurate thermal imaging measurements was added at the end of the research as an additional rationale in this field. The closer we get to room temperature, less visible is the SSE and reliability of its determination. In the future research we will concentrate on the accuracy of thermal imagers related to body temperature measurements but as said before this research was initiated before the pandemic.
The temperatures were chosen based on availability of very stable and homogeneous temperature radiation sources which cover the entire temperature range of thermal imaging cameras. Typically, the most interesting ranges are up to 250 °C (some low-cost cameras work only up to that range), then up to 500 °C (most of the cameras work in the range up to 500 °C), extended range goes either up to 1000 °C or 2000 °C. In our case we have a blackbody that operates up to 1500 °C but due to expensive heaters which should be used at the highest temperature for a short time, we performed measurements up to 1400 °C only.

  1. What’s the effect of different emissivity?

Continuing from the first answer. The knowledge of emissivity if very important before even starting a non-contact temperature measurement. Unfortunately, the emissivity is a parameter related to temperature, radiation wavelength and the angle of observation. On top of that traceability of emissivity measurements is not that rigorous as it should be, neither it is simple or low-priced, because it requires very expensive equipment and extensive knowledge and expertise. In addition, at lower emissivities other effects come into consideration which cause large technical problems for execution of measurements and add new uncertainty contributions, which are far from negligible (background radiation, reflected radiation).
Unfortunately, we do not have better calibration sources than the blackbodies, which only provide the corrections and uncertainty for the emissivity close to 1 (in the wider range of wavelengths). There are also so-called greybodies with effective emissivity of around 0.95 in the IR spectrum which are used for calibration of radiation thermometers with pre-set emissivity of 0.95. The story of calibration ends here and nobody knows how useful the corrections provided in calibration against the blackbody actually are, if later the measurement is performed at lower emissivity values. The Plank’s law cannot be applied here because there are so many elements/media/objects between the measured object and the detector (atmosphere, optical system), which change the transmission function. Determination of the latter is an intellectual property of manufacturers and are kept as business secrets, therefore we are not able to calculate the temperature for a radiation temperature instrument even if we are able to measure thermal radiation or if we would know the raw signal of the detector.
Our research in the near future will be concentrated to provide as many answers as possible related to calibration of instruments at lower emissivity values. But that is beyond the scope of this paper.

  1. Line 79: The subscript of the letters should pay more attention

The error occurred when exporting a Word document to pdf. We will pay attention when the new pdf version will be submitted.

Reviewer 2 Report

The manuscript entitled ‘Evaluation of the size-of-source effect in thermal imaging cameras’ falls within the scope of the journal Sensors. The paper contains very interesting experimental results as well as measurement procedures. The manuscript concerns the accuracy of the temperature measurement using a thermal camera (thermal analysis of the image). The article fits well in solving the current world health problems. It is of sufficient scientific interest and has originality in its technical content to merit publication. The authors have cited the relevant literature. Methods, interpretations of results are correct. The authors presented extensive material supporting the conducted research. The issues were well presented. In terms of content, the analysis does not raise any objections. The arrangement of work maintains substantive continuity and constitutes a logical whole.

However, the manuscript needs some minor corrections.

Figures 9 -18 and 22 – 28 should be enlarged.

Author Response

The manuscript entitled ‘Evaluation of the size-of-source effect in thermal imaging cameras’ falls within the scope of the journal Sensors. The paper contains very interesting experimental results as well as measurement procedures. The manuscript concerns the accuracy of the temperature measurement using a thermal camera (thermal analysis of the image). The article fits well in solving the current world health problems. It is of sufficient scientific interest and has originality in its technical content to merit publication. The authors have cited the relevant literature. Methods, interpretations of results are correct. The authors presented extensive material supporting the conducted research. The issues were well presented. In terms of content, the analysis does not raise any objections. The arrangement of work maintains substantive continuity and constitutes a logical whole.

However, the manuscript needs some minor corrections.

Figures 9 -18 and 22 – 28 should be enlarged.

Thank you for the comments. The figures were resized, while fitting into two-column paper layout.

Reviewer 3 Report

The paper is focused on the metrological evaluation of the performance of an infrared camera and, in particular, on the size of the detectable thermal source.

The work would be interesting, but it must be improved.

*Abstract

Although the abstract seems to introduce a study on thermographic evaluation for body temperature monitoring, actually the paper does not present specific data on this topic (in vivo tests).

Lines 7-10, improve the clarity of the sentences;

Line 17, improve the clarity of the sentence;

Lines 18-20, this sentence does not seem to be related to the previous discussion;

I suggest rewriting the whole abstract.

* Introduction

The introduction section presents a limited state-of-the-art. In fact, it does not address the topic of the quantitative evaluation in thermographic applications, but only the size of the detectable thermal source Furthermore, the aim of the paper is not clearly presented.

For example, I suggest reading and considering these works in the quantitative thermographic evaluation field:

https://www.sciencedirect.com/science/article/pii/S0735193309001146

https://www.sciencedirect.com/science/article/abs/pii/S0924424718305594

Lines 61-69, improve the clarity of the sentences;

Lines 84-98, cite the suitable references;

Lines 99-102, cite the suitable references;

Lines 126-159, the section has a didactic character which is not useful for paper;

Lines 160-162, define the aim of the paper only at the end of the introduction.

*Material and methods

In this section, there are some important shortcomings.

Line 177, this is not a valid reason. All infrared cameras used in the research applications are able to acquire temperatures of each individual pixel;

Lines 197-200, what do you mean with “To stabilize the camera at the operating temperature”?

Line 230, at lines 170-171 the minimum focus distance is stated to be 15 cm, but your investigation was also performed at 10 cm. The effect of the out-of-focus is extremally relevant on the accuracy of the infrared camera;

Lines 260-261, this sentence is repeated many times in the text;

Lines 272-273, this sentence is repeated many times in the text;

Lines 278-283, a font visualization error does not allow to easily understand the text.

*Results

The results are interesting, but the are not well presented. Furthermore, some figures have not a suitable comment. For example, provide an explanation for the trend in fig 14.

I suggest improving the robustness of your work by introducing a statistical analysis (standard deviation, error bars, etc.)

Fig 10, highlight that the image of the left is in visible range; provide the colour scale for the image of the right;

Figs 11, 12, 13 check the measurements at 10 cm (as previously suggested);

Lines 325-334, some font visualization errors do not allow to understand the text;

Figs 14, 18, 22, 23, 24, 26, 28, improve the quality.

In my opinion the authors must address the highlighted lacks. Hence, I suggest a major revision before the publication of the paper.

Round 2

Reviewer 1 Report

The non-contact temperature measurement is totally different from contact temperature measurement.

The radiative transfer equation is not the same for contact ones.

Reviewer 3 Report

All my comments have been addressed. In fact, the authors have sufficiently improved their paper.
In my opinion, the paper can be considerated for publication.